# Botulinum Neurotoxin Therapy for Lingual Dystonia Using an Individualized Injection Method Based on Clinical Features

**DOI:** 10.3390/toxins11010051

**Published:** 2019-01-17

**Authors:** Kazuya Yoshida

**Affiliations:** Department of Oral and Maxillofacial Surgery, National Hospital Organization, Kyoto Medical Center 1-1 Mukaihata-cho, Fukakusa, Fushimi-ku, Kyoto 612-8555, Japan; kayoshid@kyotolan.hosp.go.jp; Tel.: +81-756-419-161; Fax: +81-756-434-325

**Keywords:** lingual dystonia, botulinum toxin therapy, botulinum neurotoxin, tongue, intrinsic muscle, extrinsic muscle, oromandibular dystonia

## Abstract

Lingual dystonia is a debilitating type of oromandibular dystonia characterized by involuntary, often task-specific, contractions of the tongue muscle activated by speaking or eating. Botulinum neurotoxin (BoNT) has been used to treat lingual dystonia; however, it is known to cause serious complications, such as dysphagia and aspiration. The purpose of this study was to evaluate the efficacy and adverse effects of individualized BoNT therapy for lingual dystonia. One-hundred-and-seventy-two patients (102 females and 70 males, mean age: 46.2 years) with lingual dystonia were classified into four subtypes based on symptoms of involuntary tongue movements: protrusion (68.6%), retraction (16.9%), curling (7.6%), and laterotrusion (7.0%). Patients were treated with BoNT injection into the genioglossus and/or intrinsic muscles via individualized submandibular and/or intraoral routes. Results were compared before and after BoNT therapy. Botulinum neurotoxin was injected in 136 patients (mean: 4.8 injections). Clinical sub-scores (mastication, speech, pain, and discomfort) in a disease-specific rating scale were reduced significantly (*p* < 0.001) after administration. Comprehensive improvement after BoNT injection, assessed using the rating scale, was 77.6%. The curling type (81.9%) showed the greatest improvement, while the retraction type showed the least improvement (67.9%). Mild and transient dysphagia occurred in 12.5% of patients (3.7% of total injections) but disappeared spontaneously within several days to two weeks. No serious side effects were observed. With careful diagnosis of subtypes and a detailed understanding of lingual muscle anatomy, individualized BoNT injection into dystonic lingual muscles can be effective and safe.

## 1. Introduction

The tongue is a highly muscular organ, which plays crucial roles in speech, deglutition, respiration, mastication, and taste. It is composed of a root, an apex, a curved dorsum, and an inferior surface. It is attached to the hyoid bone, mandible, styloid processes, soft palate, and pharyngeal wall [1]. The tongue is made up of four extrinsic and four intrinsic muscles (Figure 1 and Figure 2), which are divided by a median fibrous septum: The lingual septum. The extrinsic muscles act to change the position of the tongue, while the intrinsic muscles alter the shape of the tongue [1]. The extrinsic muscles include the bilateral genioglossus, hyoglossus, styloglossus, and palatoglossus (Figure 1 and Figure 2). 

The genioglossus arises from a tendon attached to the genial tubercle. It fans out backwards and upwards. The superior fibers of the muscle ascend forwards to enter the whole length of the ventral surface of the tongue from root to apex, intermingling with the intrinsic muscles. Muscles on contralateral sides are separated by the lingual septum. The attachment to the genial tubercle prevents the tongue from obstructing breathing. The genioglossus provides traction to move the tongue forwards to protrude its apex from the mouth. The vast majority of patients with lingual dystonia show dystonic contraction in the genioglossus. The hyoglossus arises from the hyoid bone. It runs vertically up to enter the side of the tongue between the styloglossus, laterally, and the inferior longitudinal muscle, medially. The hyoglossus retracts and depresses the tongue. The styloglossus arises from the hyoid process of the temporal bone, drawing the sides of the tongue up and backwards to create a trough for deglutition. The palatoglossus arises from the palatine aponeurosis. It depresses the soft palate. This muscle elevates the dorsum of the tongue during swallowing. 

The intrinsic muscles consist of four pairs of muscles: the superior and inferior longitudinal muscles, as well as the transverse and vertical muscles. These muscles originate from and insert into the tongue, running along its length. The superior longitudinal muscle runs beneath the mucosa of the dorsum of the tongue. This muscle shortens the tongue and also turns the apex and sides upwards to make the dorsum concave. The inferior longitudinal muscle is close to the inferior lingual surface, between the genioglossus and hyoglossus. It extends from the root of the tongue to the apex and draws the apex downwards to make the dorsum convex. The transverse muscle runs laterally from the median lingual septum to the submucous fibrous tissue in the lingual margin. This muscle makes the tongue narrow and elongated. The vertical muscle passes from the dorsal to the ventral aspects of the tongue in the anterior borders. This muscle flattens and widens the tongue. Fibers of the transverse and vertical muscles partially intermingle.

Dystonia is a hyperkinetic movement disorder that is characterized by sustained or intermittent muscle contractions, resulting in abnormal repetitive movements and/or postures [2]. Oromandibular dystonia is a focal type of dystonia that involves the masticatory and/or lingual muscles. It is subdivided into jaw closing dystonia, jaw opening dystonia, lingual dystonia, jaw deviation dystonia, jaw protrusion dystonia, and a combination of these subtypes [3,4,5,6]. Dystonic contraction of the tongue can interfere with very important daily activities such as speaking, mastication, and deglutition, thus causing vocational, masticatory, esthetic, and, consequently, social disabilities. Lingual dystonia is a debilitating form of oromandibular dystonia. The involuntary lingual movements include repetitive or episodic tongue protrusion that are often induced task-specifically when speaking and/or eating [7,8,9,10,11,12,13]. Although the direct cause of lingual dystonia remains unknown, speech-induced lingual dystonia can be regarded as an occupational dystonia in certain cases [13]. Detectable secondary causes include head injury [14], electric injury [15], degenerative or inherited diseases [9,11,16], and varicella infection [17]. 

Lingual dystonia is a rare focal dystonia, with a prevalence of 4% [11]. The overall prevalence of primary dystonia was estimated at 164.3 per million [18]. The prevalence of task-specific dystonia ranges from 7 to 69 per million [19,20]. The prevalence of oromandibular dystonia was estimated to be around 68.9 per million [21]. 

The pharmacological therapy for treating lingual dystonia is unstable, and, in most cases, unsatisfactory. Muscle afferent block therapy involves local injection of diluted lidocaine and ethanol. It aims to reduce the effect of muscle spindle afferents without unfavorable weakness [22]. Kaji et al. [22,23] reported this method for the treatment of writer’s cramp and cervical dystonia. We applied this method for oromandibular dystonia [4,24,25]. 

The clinical applications of botulinum neurotoxin (BoNT) have been expanding in a variety of diseases [26,27,28,29,30], and the number of studies investigating the favorable effects of BoNT injection has increased [7,8,9,10,11,12,15,31,32,33]. Although early studies reported life-threatening complications like significant dysphagia [7], aspiration pneumonia [7], and serious swallowing and breathing difficulties [9], BoNT therapy is recognized as a feasible treatment option for this condition [11,12]. The tongue, which is likely the most difficult organ in the stomatognathic system to target with BoNT injection, has crucial functions in speaking, swallowing, mastication, and respiration that make injection riskier. Some of the potential consequences of paralyzing the tongue muscles include dysphagia, dysarthria, masticatory disturbance, and breathing difficulty. To minimize risks for these unfortunate complications, multiple routes for injection, including the submandibular route [7,8,11,12,15,32], the intraoral route [10,33], and injection into other muscles instead of the lingual muscles [34] have been attempted.

The aim of this study was to evaluate the effectiveness and complications of BoNT therapy by individualized injection for lingual dystonia.

## 2. Results

In this retrospective observational study, clinical characteristics, treatments, response, and adverse effects of BoNT therapy in four subtypes of lingua dystonia (protrusion, retraction, curling, and laterotrusion types) were analyzed based on the results observed by one physician at a single institution.

### 2.1. Clinical Characteristics

From the total of 252 patients with involuntary tongue movements, the author excluded 80 patients with lingual or orolingual dyskinesia, psychogenic (functional) movement disorders, and generalized dystonia. One hundred and seventy-two patients with lingual dystonia (102 females and 70 males, mean age ± SD: 46.2 ± 13.7 years) were analyzed in this study (Table 1). The mean duration of neuroleptic and/or tranquilizer usage was 12.5 ± 8.3 years in the 53 patients with tardive dystonia. 

All patients exhibited stereotypical contractions. Task-specificity was observed in 155 patients (90.1%) during speaking, chewing, or swallowing (Table 1). One-hundred-and-seventeen patients (68.2%) patients utilized sensory tricks such as chewing gum, candy, touching their jaws with their hand or finger, handkerchief, or a mask (Table 1). Other sensory tricks included the use of a pipe, piece of wood, straw, toothpick, and cotton. One-hundred-and-twenty-two patients (70.9%) reported morning benefit. 

### 2.2. Subtypes of Lingual Dystonia

The most prevalent type of lingual muscle contraction was protrusion (118 patients, 68.6%), followed by retraction (29 patients, 16.9%), curling (13 patients, 7.6%), and laterotrusion (12 patients, 7.0%) (Figure 3). 

### 2.3. Treatments

The author treated patients with a combination of pharmacotherapy, muscle afferent block, BoNT injection, sensory trick splint, and Myomonitor. The author prescribed trihexyphenidyl, baclofen, clonazepam, tiapride, zolpidem, as well as Chinese medicines in 87 patients (50.6%). Muscle afferent block therapy was administered a total of 709 times (mean; 7.0 ± 8.2 times, range; 1–40 times) in 102 patients (59.3%). Sensory trick splints were fabricated for the mandibular dental arch for 43 patients (25.0%). Patients were instructed to insert the splints during the daytime. Myomonitor, which applies transcutaneous electro-neural stimulation, was used for several patients with jaw elevator muscle pain. 

### 2.4. Botulinum Toxin (BoNT) Therapy

The total scores, as evaluated by using a disease-specific rating scale at baseline, were significantly (*p* < 0.001) reduced following BoNT therapy, as measured at the end of the follow-up visit (Figure 4). The average dose per injection into the lingual muscle was 43.1 ± 5.3 units. 

Botulinum neurotoxin was injected into the genioglossal muscle in 136 patients (79.1%) a total of 650 times (mean; 4.8 ± 3.9 times, range; 1–19 times) without any significant complications. The results are summarized in Table 2. 

Clinical sub-scores (mastication, speech, pain, and discomfort) were significantly reduced (*p* < 0.001) after administration of BoNT (Table 3). Comprehensive improvement following BoNT injection, as assessed by a disease-specific rating scale, was 77.6% in the protrusion type, 78.7 ± 14.2% in the retraction type, 67.9 ± 10.2% in the curling type, 81.9 ± 35.5%, and 73.2 ± 12.6% in the laterotrusion type. The curling type showed the greatest improvement while the retraction type showed the least improvement. However, this difference did not reach significance.

In total, 12.5% of the patients (3.7% of total BoNT injections) had mild and transient dysphagia that disappeared spontaneously within several days, but lasted up to two weeks in some cases. No serious complications were observed. Depending on the symptoms of each patient, injected muscles, other than extrinsic and intrinsic muscles, included the lateral pterygoid (24 patients), masseter (23 patients), posterior belly of the digastric (8 patients), medial pterygoid (5 patients), temporalis (4 patients), geniohyoid (4 patients), anterior belly of the digastric (3 patients), sternocleidomastoid (3 patients), orbicularis oris (3 patients), risorius (2 patients), mentalis (1 patient), and buccinator (1 patient) muscles.

## 3. Discussion

This study is the first to report on the efficacy and complications of the use of BoNT injection for lingual dystonia based on subtype classification. Although some previous reports have been made based on database review, this study was a report on patients diagnosed, treated, evaluated, and followed by the same oromandibular dystonia specialist. Therefore, inconsistencies in diagnosis, and inter-examiner differences, were thus kept at a minimum. 

### 3.1. Limitations of This Study

This retrospective observational study was uncontrolled in an open-label fashion. The small sample size in the laterotrusion and curling groups may have been inadequate for statistical analyses. If the patients showed tongue protrusion accompanying lateral deviation or curling, the patients were classified into the protrusion type. As a result of this, the protrusion type represented about two-thirds of the patients observed, while the laterotrusion and curling types were less represented. This low statistical power might negatively influence the likelihood that a significant finding actually reflects a true effect. Hence, to obtain significant evidence, further studies with more patients are necessary.

### 3.2. Lingual Dystonia

The extrinsic and intrinsic muscles can greatly, and accurately, alter the position and shape of the tongue, which enables coordinated movements for processes such as phonation and speech, and mastication and swallowing. Detailed knowledge of the tongue muscles and their functions is a prerequisite for safe and effective BoNT therapy for lingual dystonia.

Due to the low prevalence of lingual dystonia, it has previously been reported on in single case reports [15,26,33,35,36,37,38,39,40,41,42] or case series [7,8,9,10,11,12,13]. This study is the largest series of cases with lingual dystonia reported thus far. Particularly, isolated lingual dystonia is extremely rare. Lingual dystonia was reported at a prevalence of 4% among all types of dystonia [11]. The reported prevalence range of lingual dystonia among all types of oromandibular dystonia was between 17–27% [6,43,44]. Many patients with lingual dystonia have previously visited dentists or oral and maxillofacial surgeons. These patients are often misdiagnosed as having temporomandibular disorders, bruxism, or dental problems, or remain undiagnosed [13,43,45]. Therefore, the true prevalence of lingual dystonia is likely much higher than previously estimated.

The population of the present report differs considerably from previous studies, which mostly reported patients with general dystonia secondary to degenerative, inherited, and post-encephalitic diseases, and neuroacanthocytosis [9,11,16]. Patients with hyperkinetic involuntary movements of the tongue, particularly patients with isolated lingual dystonia, visit our department from great distances. The author has created a website for patients with involuntary movement in the stomatognathic system [46]. This website has been accessed by more than one million visitors from over 190 countries and regions [47]. At our department, the author offers a wide range of multimodal therapies, including medication, muscle afferent block therapy [4,23,25], BoNT therapy [48,49,50], sensory trick splint therapy [45], Myomonitor, and surgery [51,52]. As there are no other hospitals that specialize in oromandibular dystonia, a large number of patients are referred to the author from all over Japan and from other parts of the world as well. In addition, it is possible that many patients who had already abandoned consultation and/or treatment might have visited our clinic after informing themselves about oromandibular dystonia via the author’s website [46,47].

In this study, 90.1% of patients exhibited task-specificity (Table 1). Researchers have reported two hypotheses regarding the underlying mechanisms of focal task-specific dystonia [53,54,55,56,57,58]. The first hypothesis is impaired inhibition and abnormal plasticity regulation [53,54,55,56]. The second hypothesis proposes that task-specific dystonia results from dysregulation of plasticity in the brain [57,58]. The above-mentioned studies are based on results from patients with focal dystonia, such as writer’s cramp. We have attempted to clarify the cortical neurophysiology related to jaw movements and perception in the stomatognathic system [59,60,61,62,63,64], including the tongue [62,63] and the soft palate [61], using encephalography [59,60] and magnetoencephalography [61,62,63]. Recordings, by means of neuroimaging and noninvasive stimulation techniques, are difficult in patients with oromandibular dystonia due to artifacts from orofacial muscle activity and involuntary jaw movements. We have reported movement-related cortical activity before mandibular movements in oromandibular dystonia [64]. However, future studies are necessary to clarify the neurophysiology, pathophysiology, and etiology of task-specific lingual dystonia.

### 3.3. Injection Method of BoNT for Lingual Dystonia

The success rate of BoNT therapy, as well as the incidence of injury to the adjacent tissues, such as arteries or nerves, is closely related to the accuracy of the needle placement [48,49,50]. Botulinum neurotoxin injection into the lingual muscle occasionally causes serious dysphagia [7], aspiration pneumonia [7], and swallowing and breathing difficulties [9]. Many authors have reported several methods for BoNT injection. Most of the researchers reported using the submandibular approach [7,8,11,12,15,32], while intraoral approaches, including the superficial approach, were also reported [10,32]. In the submandibular method, injection is performed in one or two sites, bilaterally. In another approach, BoNT injection into the masseter or the lateral pterygoid muscles was performed instead of injection into the tongue muscle [34]. 

There are various patterns of involuntary tongue contraction, such as protrusion or laterotrusion. Furthermore, the extent of the dystonic contraction shows large interindividual differences. In light of this variability, the same method of approach is unlikely to be effective for all patients. In spite of this, researchers used the same injection method for all patients. In this study, the pattern of muscle contraction varied from patient to patient. The author classified the pattern into four types, with variations within types. The author determined the total amount of BoNT and the ratio of the toxin injected into each site and observed its effects on contraction type and the severity of muscle contraction, based on electromyography (EMG) examination. This individualized injection method could maximize response and minimize complications. The more precisely the tip of needle is placed in the dystonic muscles, the more likely the amelioration of patient symptoms and the lower the risk of adverse events [48,49]. 

The injections should be conducted by a physician or an oral and maxillofacial surgeon extensively trained in the anatomy and physiology of the stomatognathic system and in an environment equipped to manage potential severe complications. Diagnosis, treatment, and research on oromandibular dystonia have been conducted mainly by neurologists and partly by otolaryngologists or oral and maxillofacial surgeons. However, the specialists of the stomatognathic system are oral and maxillofacial surgeons, who execute daily operations, and diagnose and treat a variety of diseases in the stomatognathic system. Most patients with masticatory disturbance, objectively, seem to witness considerable improvement upon treatment; however, subjective responses did not always reveal favorable results. Masticatory function can be evaluated only by oral and maxillofacial surgeons. They have detailed knowledge concerning the tongue compared to other medical experts. A multidisciplinary team approach, including a neurologist, an oral and maxillofacial surgeon, a neurosurgeon, and an otolaryngologist may be required for proper diagnosis and treatment of lingual dystonia [45].

The only adverse event that occurred in this study was mild and transient dysphagia, which disappeared spontaneously after several days to two weeks. It is reasonable to postulate that the dysphagia could be related to imbalances in tongue muscles caused by BoNT injection, as muscular imbalance could hamper normal swallowing. One researcher reported lower rates of dysphagia (2.2% per total injections) [11] than those observed in this study (3.7%). This could be related to the higher doses of BoNT used in this study (43 units) compared to that used in the other study (26.6 units) [11]. The doses of BoNT were adjusted if the patient experienced dysphagia. 

### 3.4. Responses in Each Subtype

So far, the contraction patterns of lingual dystonia have not been properly studied. In this study, the most prevalent type of lingual muscle contraction was protrusion (68.6%), followed by the retraction (16.9%), curling (7.6%), and laterotrusion (7.0%). These results might be due to the fact that the patients with a tendency to develop orolingual dyskinesia were excluded in the analysis. The author attempted to evaluate the differences in each type; however, the sample sizes in the laterotrusion and curling groups were insufficient for reliable statistical analyses. 

The protrusion type is prevalent in lingual dystonia, sometimes even accompanying laterotrusion and curling (up or down) tongue contractions. Botulinum neurotoxin should be injected mainly into the genioglossus. When using the submandibular route, the needle penetrates the geniohyoid, mylohyoid, and digastric muscles, which could result in dysphagia; therefore, the mandibular approach should be performed at one site, bilaterally. 

The retraction type is the most difficult subtype of lingual dystonia to diagnose. Since the patients cannot reproduce dystonic contraction during EMG examination, it is often difficult to detect the true muscles causing the dystonic symptoms. The muscles can include a wide range of tongue musculature such as the genioglossus, the intrinsic muscles, the geniohyoid, the hyoglossus, and the digastric muscles. EMG examination must be carefully performed. There was little improvement, and complications were higher, in this subtype than in any others. Before BoNT administration, we often tried muscle afferent block into the targeted muscles suspected of causing symptoms [24]. These nerve blocks led to transitory relieve of symptoms if procedures were successful. However, those patients with the retraction type did not respond as well compared with other subtypes. Therefore, the ratio of BoNT therapy was lower (12 out of 29 patients, 41.3%) in this subtype than in other subtypes (protrusion, 89.8%; curling, 76.9%; laterotrusion, 66.7%). 

The laterotrusion and curling types, without tongue protrusion, have dystonic contraction in limited portions of the tongue. If BoNT can be precisely injected into these contracting areas, patients’ symptoms can be relieved, dramatically. The curling type showed the greatest improvement upon BoNT injection; however, statistical significance was not reached. 

Due to large inter-individual differences in the clinical features of dystonia, the individualized method described in this paper is likely to produce more satisfactory outcomes. All patients responded to the treatment, and no patients remained unchanged or worsened. To elucidate the difference in responses among the four subtypes, further studies with larger samples are required.

### 3.5. Rating Scale for Oromandibular Dystonia

Many rating scales or questionnaires have been used to evaluate the various types of dystonia; however, only few scales have been assessed for clinical use [65,66,67,68]. There have been no objective or comprehensive rating scales for evaluating the therapeutic effects of treatments for oromandibular dystonia. Moreover, patients with this condition often have a variety of complaints because the stomatognathic system plays important roles in mastication, speech, swallowing, and expression. In fact, the symptoms and complaints of patients may vary much more from patient-to-patient than between types of focal dystonia. In 2002, we reported a simple rating scale for oromandibular dystonia, and evaluated 44 patients using this scale [24]. In this study, to rate changes in patient symptoms more comprehensively, the author used this clinical scoring system, based on the sub-scores for pain, mastication, speech, and discomfort (Table 4). A disease-specific, validated questionnaire was developed, in which psychosocial and cosmetic scores were added for measuring the quality of life in oromandibular dystonia patients [68]. The questionnaire was previously used to evaluate quality of life after BoNT therapy for lingual dystonia in a recent study [12]. The author developed a disease-specific rating scale for oromandibular dystonia (oromandibular dystonia rating scale: OMDRS), which combined an examiner-rated scale and a patient-rated questionnaire. The rating scale included eight subscales: mastication, swallowing, speech, pain, discomfort, cosmetic, activities of daily living, and psychosocial. The author collected data from patients with oromandibular dystonia and validated the oromandibular dystonia rating scale. The full data obtained from the rating scale cannot be reported here due to limited space, and therefore, will be published elsewhere.

### 3.6. Other Treatment Methods

In addition to BoNT injection, the author treated patients with pharmacotherapy, muscle afferent blocks [4,24,25], sensory trick splints [45], and Myomonitor. Sensory trick splints can be fabricated and adjusted only by dentists or oral and maxillofacial surgeons. Myomonitor is a very common device used for transcutaneous electro-neural stimulation in temporomandibular disorders. It has been utilized to treat several patients with jaw elevator muscle pain. A comprehensive range of multimodal treatment options must be implemented for lingual dystonia and oromandibular dystonia [45].

Muscle afferent block therapy was conducted a total of 709 times in the current study. Injection of procaine into the triceps brachii of the decerebrate cat almost abolished muscular rigidity without changing its response to electric stimulation of the brachial plexus [69]. Procaine injections into muscle have been shown to eliminate rigidity while maintaining the muscle power [70]. Muscle afferent block is used to reduce the effectiveness of the muscle spindle afferents without causing unfavorable weakness [22]. Muscle afferent block has been used to treat writer’s cramp [22,23] and spasticity [71], as well as oromandibular dystonia [4,24,25]. The therapy quickly has apparent effect just a few minutes after the injection; however, it only lasts for a short duration. After carefully repeated injections, the effects gradually last longer, and eventually, the effects could last for six months or more [24]. The effect of this treatment is mediated by the blockade of either muscle afferents or gamma motor efferents [72,73]. The T-reflex of the hand muscles was attenuated whilst their power was retained after injection of lidocaine, and the muscle spindle afferents, or gamma motor efferents, were hypothesized to be blocked by muscle afferent block [22]. The mean response of the jaw elevator muscles to muscle afferent block (70%) was significantly higher than that of the depressor muscles (38%) [4]. It was postulated that the different numbers of muscle spindles innervating these muscles were responsible for these results [4]. Therefore, muscle afferent block therapy is indicated for jaw elevator muscles (the masseter, temporalis, and medial pterygoid muscles) with rich muscle spindles, but not for facial muscles, which contain no muscle spindles. Repeated injections of lidocaine are required for blockade of jaw depressor muscles (inferior head of the lateral pterygoid muscle), which contain fewer muscle spindles than the jaw elevator muscles. We achieved improvement by repeated injections, performed accurately, using a customized EMG needle insertion guide into the lateral pterygoid muscle, without major adverse effects. This is comparable to injection into the elevator muscle, although the number of injections was significantly larger [74]. Muscle afferent block is less effective than BoNT therapy, and thus requires more injections. However, muscle afferent block is safer and less expensive than BoNT, and may be very helpful for patients who cannot afford expensive BoNT therapy. We often used muscle afferent block to predict the effect of BoNT therapy. If muscle afferent block can relieve patients’ symptoms, even temporarily, there is a high probability they will respond to BoNT. However, in cases where there is no benefit with afferent block, BoNT is unlikely to help. Muscle afferent block could also prolong the effectiveness of BoNT between injections. Furthermore, it can be valuable for patients who developed antibodies to BoNT. 

## 4. Conclusions

With understanding of both the muscle anatomy and contracture pattern of the lingual muscles, individualized BoNT injection into these muscles is expected to be effective and safe for the treatment of lingual dystonia. 

## 5. Methods and Materials

### 5.1. Patients

Two-hundred-and-fifty-two patients (159 females and 93 males, mean age ± standard deviation [SD]: 52.5 ± 17.4 years) with involuntary tongue muscle contractions visited our department from 2007 to 2018. Patients who were suspected of having a degenerative, inherited, or other neurological diseases were referred to neurologists in our clinic. Patients who had already visited neurologists or neurosurgeons prior to presentation at our clinic were neurologically examined to ensure no abnormal findings were present. Patients with orolingual dyskinesia, psychogenic movement disorders, and generalized dystonia were excluded from analysis. Diagnosis of lingual dystonia was established by electromyography and characteristic clinical features of focal dystonia [13,45]. These clinical features included stereotypy, task-specificity, morning benefit (the tendency of dystonia to show milder symptoms in the morning), and co-contraction. Sensory tricks are various voluntary maneuvers that ameliorate dystonic symptoms [13,45].

All patients involved in this study provided written informed consent after receiving a full explanation of the planned treatment and possibility of publication of results. This study was performed in accordance with the tenets of the Declaration of Helsinki and was approved by the institutional review board and ethics committee of Kyoto Medical Center (approval date: 21 September 2007; approval code: 2007-184).

Phenomenology and clinical characteristics were evaluated in the 172 patients diagnosed with lingual dystonia (102 females and 70 males, mean age ± SD: 46.2 ± 13.7 years). The patients’ chief complaints were dysarthria, masticatory disturbance, dysphagia, discomfort, tongue pain, and aesthetic disorder. All of the patients had stereotypic contraction of the tongue. The demographic characteristics of the patients with lingual dystonia are summarized in Table 1. Fifty-three (30.8%) patients (36 females and 17 males, mean age ± SD: 47.5 ± 12.7 years) had been prescribed psychiatric medication and therefore had tardive dystonia (Table 1).

The associated subtypes of oromandibular dystonia in our cohort were jaw-opening dystonia (19 out of 172 patients, 11.0%), jaw-closing dystonia (10 patients, 5.8%), jaw-deviation dystonia (4 patients, 2.3%), jaw-protrusion dystonia (1 patient, 0.6%), and lip dystonia (1 patient, 0.6%) (Table 1). Associated movement disorders included writer’s cramp (7 patients, 4.1%), blepharospasm (9 patients, 5.2%), cervical dystonia (6 patients, 3.5%), embouchure dystonia (2 patients, 1.2%), and spasmodic dysphonia (2 patients, 1.2%) (Table 1).

### 5.2. Classification of Lingual Dystonia into Four Subtypes

After the exclusion of patients with orolingual dyskinesia, psychogenic movement disorders, and generalized dystonia from the original 252 patients who presented at our clinic, patients with lingual dystonia were further evaluated. The pattern of lingual muscle contraction was classified into four subtypes based on the phenomenology of dystonic contraction: protrusion, retraction, curling, and laterotrusion (Figure 1). The protrusion type was characterized by lingual protrusion, usually out of the mouth. In the retraction type, the tongue was not protruded but retracted, or there was contraction of the tongue’s base or the whole tongue. The laterotrusion type presented as lateral deviation of the tongue. The curling type was characterized, not by a dyskinetic movement, but by an upward curling contraction of the tongue, which was often task-specific. Several patients with the protrusion type of lingual dystonia also presented with the laterotrusion or curling type of contraction. Such patients were placed into the protrusion group if the tongue protruded out of the mouth. 

### 5.3. Treatment

After careful examination, anticholinergic or antispasmodic medications were prescribed for mild to moderate cases. For patients with tardive dystonia, who were referred by a psychiatrist or a neurologist, the author performed blocking of the involuntary muscle contractions by intramuscular injection if the patient reported not taking other oral medications or stated unresponsiveness to pharmacotherapy prescribed by the referring physicians, or the response to oral medication use was unsatisfactory. First, muscle afferent block therapy [4,24,25] was performed on hyperactive muscles. Treatment was continued if results were favorable and transient BoNT therapy was not attempted. However, BoNT therapy was conducted for patients with severe muscle contractions. If the muscle afferent block yielded unfavorable responses, a sensory trick splint was indicated [45]. If splint therapy or a BoNT injection was only partially effective, psychiatric or neurological therapies were attempted. Surgical intervention (coronoidotomy) [51,52] was performed for several patients with severely limited mouth opening. Finally, non-responders to all therapies were referred to neurologists to consider neurosurgical procedures, such as stereotactic surgery or deep brain stimulation [45].

### 5.4. Botulinum Toxin (BoNT) Therapy

The author determined the target muscles for BoNT injection based on patient symptoms and the results of EMG examination. Depending on the symptoms presented, the muscles examined included the tongue muscles (extrinsic and intrinsic muscles), as well as the digastric (anterior and posterior bellies), masseter, temporal, lateral pterygoid, medial pterygoid, geniohyoid, orbicularis oris, buccinator, mentalis, and sternocleidomastoid muscles. 

The author used 50 units vial of BoNT (Botox^®^, Allergan Pharmaceuticals, Irvine, CA, USA) for the first injection. The BoNT was reconstituted with normal saline to reach a concentration of 2.5–5 units/0.1 mL. The author began with injection doses of 15–20 units and then titrated up to higher doses to optimize efficacy while minimizing side effects. Before intraoral injection, patients were asked a gargle with a 50-fold diluted solution of Neostelin Green 0.2% mouthwash solution (Nippon Shika Yakuhin, Yamaguchi, Japan). Before percutaneous injection, the skin was disinfected with alcohol. A disposable hypodermic needle electrode (TECA™ MyoJect™ Luer Lock, 37 mm × 27 G, Natus Manufacturing Limited, Gort, Co. Galway, Ireland) was inserted into the targeted muscles. No local anesthesia was needed since patients tolerated the procedure. Correct placement of the electrode tip was verified using activity recorded by an EMG instrument (Neuropack n1, MEM-8301, Nihon Kohden, Tokyo, Japan) during voluntary dystonic movements. The recordings were amplified, filtered (low-cut filter, 10 Hz; high-cut filter, 3 kHz), and then digitized with a sample frequency of 10 kHz and 16-bit resolution. Subsequently, after aspiration, appropriate doses of the toxin were injected into the target muscles using the hypodermic needle electrode. The interval between injections was over three months. The injections were repeated after the effects diminished. Injections will be been continued, until the patients have satisfactory therapeutic effects. 

### 5.5. Individualized Injection Method for Each Subtype

There are considerable inter-individual differences and variations of lingual muscles. Figure 1 and Figure 2 show the standard anatomy of the muscles. Some lingual muscles intermingle with each other. Approximate regions are indicated with stippled circles in Figure 1 and Figure 2. Computed tomography, if applicable, would provide important information about individual anatomy. Injection methods for each subtype are summarized in Table 4. 

#### 5.5.1. Protrusion Type

Appropriate doses of BoNT (15–60 units) were determined based on EMG examination. Approximately 50–100% of the total doses were injected into the bilateral genioglossus percutaneously through submandibular region (Figure 5). Prior to injection, the patient was placed in the supine position with the head tilted backwards on the dental chair. The insertion points were defined as two sites 25–30 mm posterior from the midline of the body of the mandible and 15–20 mm apart from each other (Figure 5). 

If tongue protrusion occurred while simultaneously curling up or down, the remaining doses were injected into the superior longitudinal muscle (5-mm depth; injection to counteract tongue protrusion) or into the inferior longitudinal muscle (10–15 mm-depth; injection to counteract curling), it is easier to inject the inferior longitudinal muscles from the inferior surface of the tongue (5 mm in depth) (Figure 6). The dose of BoNT used was greater on the opposite side than on the deviated side. With this in mind, BoNT was administered into the superior and inferior longitudinal muscles on the deviated side. If the tongue showed flattening or narrowing, the remaining BoNT was injected into the bilateral vertical muscle (10 mm in depth; to counteract protrusion) and into the bilateral transverse muscle (10 mm in depth; to counteract curling). 

#### 5.5.2. Retraction Type

For this type, the author determined target muscles after careful EMG examination. These muscles included a wide range of tongue muscles that underwent contraction, such as genioglossus, intrinsic muscles, geniohyoid, and hyoglossus. Appropriate doses of BoNT were in the range of 15–50 units.

#### 5.5.3. Laterotrusion Type

The appropriate dose of BoNT was between 10 and 40 units. The BoNT was injected into the superior (5 mm in depth) and inferior (10–15 mm in depth) longitudinal muscles on the deviated side (Figure 6B). The inferior longitudinal muscle is more accessible from the inferior aspect of the tongue (5 mm in depth) than from the dorsum. If the genioglossus on the opposite side exhibited recruitment upon EMG examination, additional injection into the muscle was required.

#### 5.5.4. Curling Type

The appropriate dose of BoNT was between 10 and 40 units. The BoNT was injected bilaterally in two or three sites, from the dorsum of the tongue (Figure 6C), about 5 mm in depth, into the superior longitudinal muscle. Botulinum neurotoxin was injected into the superior longitudinal muscle near the apex if there was curling up of the apex (Figure 5).

### 5.6. Evaluation of Effect

At follow-up visits, patients were interviewed about the occurrence and the duration of clinical response and adverse effects. Clinical features, and their alteration, were documented by means of video recording at baseline and at every visit. To objectively, or comprehensively, evaluate the severity of the patients’ symptoms, the following four items were assessed separately: masticatory disturbance, dysarthria, discomfort, and pain (Table 5). Pain intensity was examined using a 100-mm visual analog scale, the endpoints of which represented “no pain” and “the worst conceivable pain”, respectively (Table 5). Other problems with discomfort, including aesthetic issues, limited mouth-opening, involuntary movements, and related social disabilities were evaluated using a scoring system for discomfort (Table 5). Each of the four items was scored on a 5-point scale from 0 (normal) to 4 (severe symptoms). The total of the four scores, ranging from 0 (normal) to 16 (severely disabled), was used as an objective measure of each patient’s condition [24]. Objective improvement (%) was calculated according to the following formula: (pretreatment scale—posttreatment scale)/pretreatment scale (%), where 0% indicated no improvement and 100% indicated complete recovery [24]. Subjective improvement was assessed by each patient on a linear, self-rating scale, ranging from 0 points (no improvement) to 100 points (complete recovery). Comprehensive and subjective improvements were evaluated one month after the last BoNT injection.

The author used a clinical scoring system involving four parameters (mastication, speech, pain, and discomfort) to objectively evaluate changes in symptoms according to clinical presentation. The total of the four sub-scores (0–4 points) was estimated as the total objective score (0–16 points).

### 5.7. Statistical Analyses

Results were evaluated before and after BoNT therapy and comparisons were made between subtypes. Differences in parameters assessed by the rating scale were evaluated at baseline and at the end of the follow-up. One-way analysis of variance was performed to compare the differences between the four subtypes. Post-hoc Scheffé tests were used to evaluate the differences. All analyses were performed using the statistical software package SPSS for Windows version 14.0 (SPSS Japan Inc., Tokyo, Japan). 

## Figures and Tables

**Figure 1 toxins-11-00051-f001:**
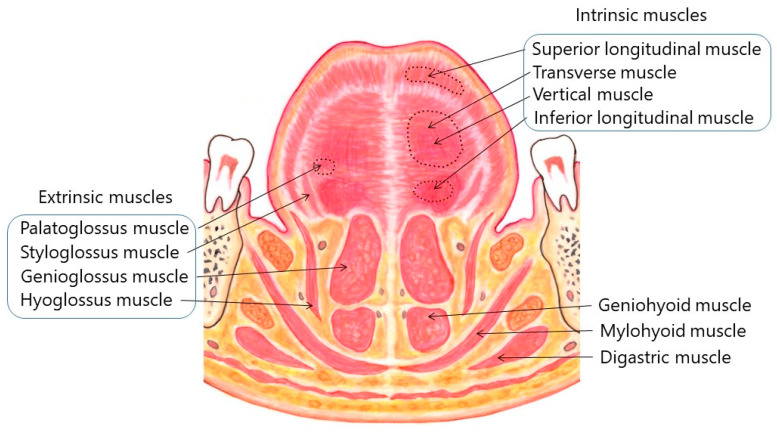
Coronal section of the tongue, viewed through the tongue, mouth, and body of the mandible opposite the lower first molar.

**Figure 2 toxins-11-00051-f002:**
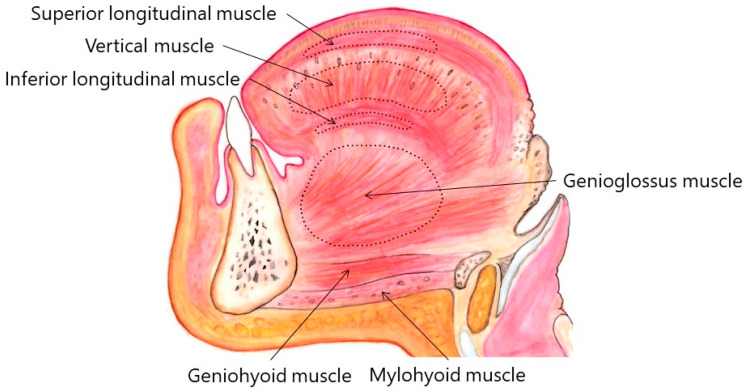
Sagittal section of the tongue. The right side of the tongue viewed from the medial side.

**Figure 3 toxins-11-00051-f003:**
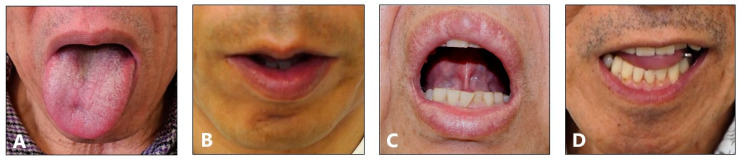
Frontal views of each subtype of lingual dystonia. Lingual dystonia was classified into 4 subtypes according to clinical symptoms; protrusion (**A**), retraction (**B**), curling (**C**), and laterotrusion (**D**) types.

**Figure 4 toxins-11-00051-f004:**
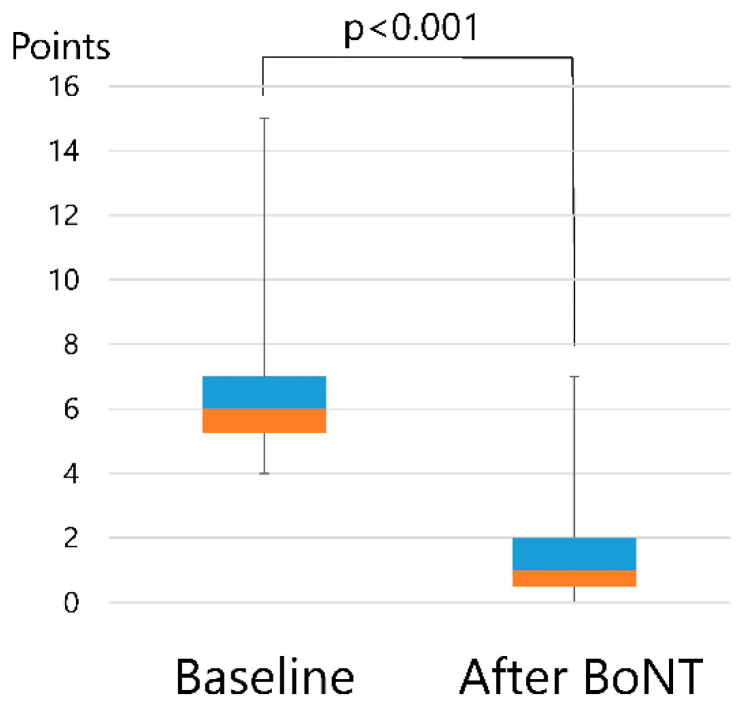
Comparison of mean total score, as measured by a disease-specific rating scale, before and after botulinum neurotoxin (BoNT) therapy. Mean total score (6.9 ± 2.4) decreased significantly after BoNT administration (1.6 ± 1.3).

**Figure 5 toxins-11-00051-f005:**
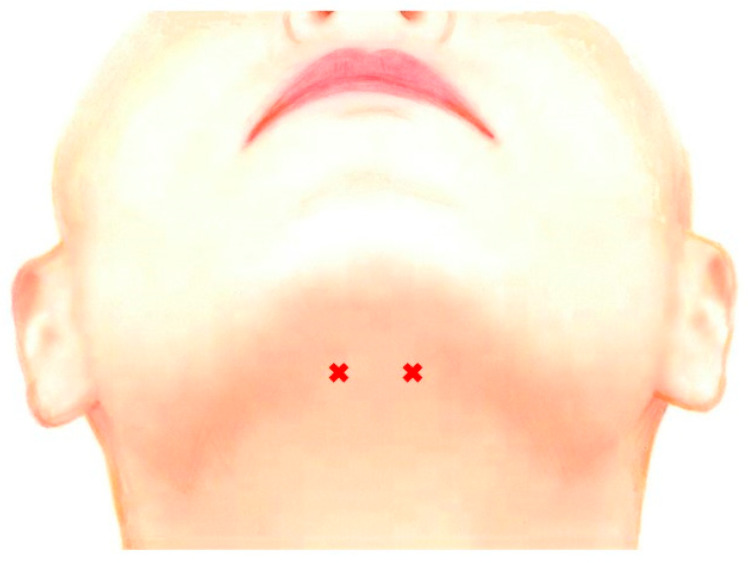
Submandibular sites for BoNT injection. The insertion points were defined as two sites 25–30 mm posterior from the midline of the body of the mandible and 15–20 mm apart from each other.

**Figure 6 toxins-11-00051-f006:**
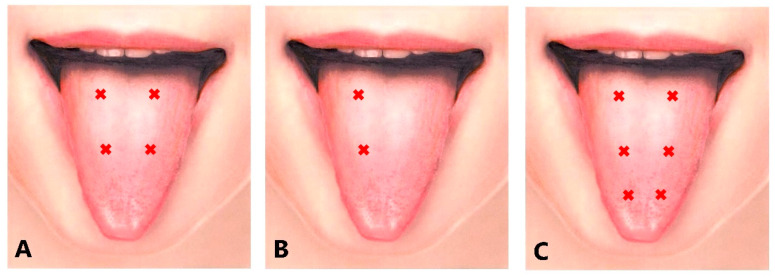
Intraoral sites of BoNT injection. Intraoral sites of the dorsum of the tongue (**A**), for laterotrusion type (right deviation) (**B**), and for curling type (**C**).

**Table 1 toxins-11-00051-t001:** Patients’ demographic characteristics.

Subtypes	Protrusion	Retraction	Curling	Laterotrusion	Total
Number of patients (*N* (%))	118 (68.6%)	29 (16.9%)	13 (7.6%)	12 (7.0%)	172 (100%)
Age (years) (mean (SD))	45.6 (13.8)	45.8 (13.4)	51.2 (15.3)	47.67 (11.2)	46.2 (13.7)
Sex (female, male) (*N* (%))	68 (57.6%) 50 (42.4%)	17 (58.6%) 11 (41.4%)	7 (53.8%) 6 (46.2%)	10 (83.3%) 2 (16.7%)	102 (59.3%) 70 (40.7%)
Duration of symptom (years) (mean (SD))	2.3 (2.6)	2.4 (2.1)	4.3 (4.2)	1.8 (1.2)	2.4 (2.6)
Tardive dystonia (*N* (%))	30 (25.4%)	11 (37.8%)	8 (61.5%)	4 (33.3%)	53 (30.8%)
Associated movement disorders (*N* (%))					
Blepharospasm	5 (2.3%)	0	3 (23.1%)	1 (8.3%)	9 (5.2%)
Writer’s cramp	3 (2.5%)	2 (6.9%)	1 (7.7%)	1 (8.3%)	7 (4.1%)
Cervical dystonia	2 (2.3%)	1 (4.0%)	2 (15.4%)	0	6 (3.5%)
Embouchure dystonia	2 (1.7%)	0	0	0	2 (1.2%)
Spasmodic dysphonia	2 (1.7%)	0	0	0	2 (1.2%)
Subtype of oromandibular dystonia (*N* (%))					
Jaw-opening dystonia	13 (11.0%)	4 (13.8.0%)	0	2 (16.7%)	19 (11.0%)
Jaw-closing dystonia	4 (3.4%)	3 (10.3%)	3 (23.1%)	1 (8.3%)	10 (5.8%)
Jaw-deviation dystonia	2 (1.7%)	0	0	2 (16.7%)	4 (2.3%)
Jaw-protrusion dystonia	1 (0.8%)	0	0	0	1 (0.6%)
Lip dystonia	1 (0.8%)	0	0	0	1 (0.6%)
Stereotypy (*N* (%))	118 (100%)	29 (100%)	13 (100%)	12 (100%)	172 (100%)
Task-specificity (*N* (%))	113 (95.8%)	28 (96.6%)	6 (46.2%)	8 (66.7%)	155 (90.1%)
Sensory tricks (*N* (%))	85 (72.0%)	17 (58.6%)	9 (69.2%)	6 (50.0%)	117 (68.2%)
Morning benefit (*N* (%))	92 (78.0%)	22 (75.9%)	4 (30.8%)	4 (33.3%)	122 (70.9%)

SD, standard deviation; *N*, count.

**Table 2 toxins-11-00051-t002:** Results of BoNT therapy in the four subtypes of lingua dystonia.

Subtypes		Protrusion	Retraction	Curling	Laterotrusion	Total
Number of patients		106	12	10	8	136
Age (years) (mean (SD))		45.6 (13.6)	44.5 (11.2)	54.4 (16.2)	51.8 (7.9)	46.5 (13.5)
Sex (female, male) (*N* (%))		58 (54.7%) 48 (45.3%)	6 (50.0%) 6 (50.0%)	6 (60.0%) 4 (40.0%)	6 (75.0%) 2 (25.0%)	74 (54.4%) 62 (45.6%),
BoNT injection (times) (mean (SD))		4.9 (3.5)	2.5 (2.0)	5.6 (7.1)	6.3 (4.7)	4.8 (3.9)
Comprehensive improvement (%) (mean (SD))		78.7 (14.2)	67.9 (10.2)	81.9 (35.5)	73.2 (12.6)	77.6 (16.7)
Subjective improvement (%) (mean (SD))		79.3 (13.5)	74.2 (9.0)	85.0 (36.8)	76.3 (11.6)	79.0 (16.0)
Adverse effects	per patient (%) (mean (SD))	14 (13.2%)	2 (16.7%)	0	1 (12.5%)	17 (12.5%)
	per session (%) (mean (SD))	21 (4.1%)	2 (6.7%)	0	1 (2.0%)	24 (3.7%)

**Table 3 toxins-11-00051-t003:** Changes in total and sub-scores in the rating scale at baseline and after BoNT therapy.

Scores	Baseline	After BoNT Therapy	*p*-Value
Mastication (points) (mean (SD))	0.81 (0.97)	0.25 (0.53)	*p* < 0.001
Speech (points) (mean (SD))	2.8 (0.81)	0.73 (0.51)	*p* < 0.001
Pain (points) (mean (SD))	0.69 (1.0)	0.1 (0.32)	*p* < 0.001
Discomfort (points) (mean (SD))	2.51 (0.72)	0.49 (0.54)	*p* < 0.001
Total (points) (mean (SD))	6.9 (2.4)	1.6 (1.3)	*p* < 0.001

**Table 4 toxins-11-00051-t004:** Summary of injection method for each subtype.

Subtypes	Protrusion		Retraction	Curling	Laterotrusion
Doses [units]	15–60		15–50	10–40	10–40
Main muscles	Bilateral genioglossus muscles (50–100% of total dose)		Bilateral genioglossus muscles (30–70% of total dose)	Bilateral superior longitudinal muscles (100% of total dose)	Superior and inferior longitudinal muscles on the deviated side (70–100% of total dose)
Additional muscles	With laterotrusion	Superior and inferior longitudinal muscles on the deviated side	Contracted muscles based on electromyography (EMG) examination including intrinsic and geniohyoid muscles	-	Genioglossus muscle on the opposite side of deviation
	With curling	Bilateral superior longitudinal muscles			
	With flattening	Bilateral vertical muscles			
	With elongation	Bilateral transverse muscles			

**Table 5 toxins-11-00051-t005:** Rating scale used to comprehensively evaluate oromandibular dystonia.

Points	Mastication Scale	Speech Scale	Pain Scale	Discomfort Scale
4	Only able to consume liquids	Inaudible (more than 50% of speech)	Severe pain (visual analog scale score: >75%)	Severe discomfort
3	Finds it difficult and takes a long time to eat soft food	Inaudible (less than 50% of speech)	Moderate, intermittent to continuous pain (visual analog scale score: 50–75%)	Moderate to severe discomfort
2	Only able to eat soft food	Audible, but difficult to comprehend	Mild continuous to moderate intermittent pain (visual analog scale score: 25–50%)	Mild to moderate discomfort
1	Able to eat anything, but it takes a long time	Finds it hard to speak clearly	Mild, intermittent pain (visual analog scale score: <25%)	Mild discomfort
0	Normal	Normal	No pain	No discomfort

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
