# Peer review of "Botulinum Neurotoxin Therapy for Lingual Dystonia Using an Individualized Injection Method Based on Clinical Features"

_toxins, 2019, doi:10.3390/toxins11010051_

Round 1
Reviewer 1 Report
The ms describes BoNT therapy for lingual dystonia using individual injection method based on clinical features.
It seemingly is a study from one physician taking care of a lot of patients from the diagnosis to the performance of various therapeutic approaches, one of it is the BoNT injection.
The manuscript - although the informationss given are really interesting - is not well prepaired.
- Information given is not correctly put into the wright capters, i.e. the anatomy of tongue muscles (taken from a textbook of anatomy) should be placed in introduction or material and methods, because is has nothing to do with the discussion.
-The chapter 3.3. should be placed in Introduction or can be omitted.
- Chapter 3.4. is redundant. Lines 244 - 256 have nothing to do with the scientific statement of the ms.
- A main point of the study - the individualized injection method - is not adaquately and reasonably decribed , it is not transparent for the reader how this was exactly done in the individual patient (dosage, times of injections, time table of injections, which muscles were injected additionally o the genioglossus muscle, and why this was done.) Why and how offen were musticatory muscles injected (see line 124).
- What was the exact value of electromyography, as it was said that is was difficult.
- There must be a clear decription in the introduction which kind of BoNT injections have been made up to now unsing which BoNT in which concentrations following which time lien in exactly which muscles (or regons) of teh tongue.
- What is indicated with the stippled line in Figs 3 and 4?
- The Figs 3 and 4 are a bit too simple for a TOXINS paper.
Author Response
- Information given is not correctly put into the wright capters, i.e. the anatomy of tongue muscles (taken from a textbook of anatomy) should be placed in introduction or material and methods, because is has nothing to do with the discussion.
Thank you very much for the comment. I replaced explanation concerning the anatomy of tongue muscles. It must be more easily accessible for readers than that in Discussion.
-The chapter 3.3. should be placed in Introduction or can be omitted.
Thank you for the comment. I considerably reduced the chapter. However, discussion on lingual dystonia is one of the most important parts of this study. Therefore, I concisely described lingual dystonia in Introduction, and explained and discussed in detailed in Discussion.
- Chapter 3.4. is redundant. Lines 244 - 256 have nothing to do with the scientific statement of the ms.
I considerably reduced the chapter. The lines 253-256 (lines ?-? in the revised manuscript) is discussion on adverse effects of BoNT terapy. I am afraid that the discussion seems to be indispensable in this study.
- A main point of the study - the individualized injection method - is not adaquately and reasonably decribed , it is not transparent for the reader how this was exactly done in the individual patient (dosage, times of injections, time table of injections, which muscles were injected additionally o the genioglossus muscle, and why this was done.) Why and how offen were musticatory muscles injected (see line 124).
The method described in this study is really individualized depending on patients’ clinical features and responses to the treatment. I added explanation in the text. However, I am afraid that further individualized detail might be too complex and confusing for the readers.
Normally I used 50 units vial of Botox. I added explanation on dosage in the text. Times and time table of injections were variable from patient to patient. Interval between injections must be over three months. Injections have continued, until the patients became satisfactory with the therapeutic effects. I added the sentences in Methods.
Based on results of EMG examination, injections were added into non-lingual muscles. I calculated results of non-lingual muscle injections. I added the following sentence in the Results. “Depending on the symptoms of each patient, injected muscles, other than extrinsic and intrinsic muscles, included the lateral pterygoid (24 patients), masseter (23 patients), posterior belly of the digastric (8 patients), medial pterygoid (5 patients), temporalis (4 patients), geniohyoid (4 patients), anterior belly of the digastric (3 patients), sternocleidomastoid (3 patients), orbicularis oris (3 patients), risorius (2 patients), mentalis (1 patient), and buccinator (1 patient) muscles.”
- What was the exact value of electromyography, as it was said that is was difficult.
I did not record the exact value of electromyography such as amplitude. If a patient can reproduce dystonic tongue movement or maintain dystonic contraction during EMG examination, and a vigorous EMG activity can be observed, it is easy to verify the correct placement of the needle in the target muscle. In some patients, it is difficult to reproduce the dystonic contraction during EMG examination. In such cases, it is difficult to verify the correct needle placement using EMG. I discussed the issue in the Discussion.
- There must be a clear decription in the introduction which kind of BoNT injections have been made up to now unsing which BoNT in which concentrations following which time lien in exactly which muscles (or regons) of teh tongue.
Thank you for the comment. Unfortunately vast majority of previous studies were single case reports or case series with small sample sizes. Clear description on injections is lacking or inadequate. I concisely explained them in Introduction, and discussed them in detail in Discussion.
- What is indicated with the stippled line in Figs 3 and 4?
Some tongue muscles fibers intermingle each other. Therefore, the boundary of other muscles are relatively unclear. I meant with the stippled lines appropriate position. I added this explanation in the figure legends.
- The Figs 3 and 4 are a bit too simple for a TOXINS paper.
Tongue muscles show considerable interindividual differences and variations. Therefore, I drew the Figs 1 and 2 (revised from Figs 3 and 4) based on some text books of anatomy. I think the figures represent standard tongue muscles and related other tissues.
Reviewer 2 Report
Overall a well written manuscript and I recommend publication but with revisions. For the benefit of the readers and to improve the manuscript, the authors should clearly describe for each type of lingual dystonia how the main muscle was approached (e.g what depth is the genioglossus on average when submandibular approach was used and what confirmatory test was used to be certain the needle electrode was in the genioglossus and not the geniohyoid). This is not clear in the methodology section for all the 4 sub-types of lingual dystonia. Without this information this manuscript is not very useful to clinicians.
Author Response
Overall a well written manuscript and I recommend publication but with revisions. For the benefit of the readers and to improve the manuscript, the authors should clearly describe for each type of lingual dystonia how the main muscle was approached (e.g what depth is the genioglossus on average when submandibular approach was used and what confirmatory test was used to be certain the needle electrode was in the genioglossus and not the geniohyoid). This is not clear in the methodology section for all the 4 sub-types of lingual dystonia. Without this information this manuscript is not very useful to clinicians.
Thank you very much for valuable comments. It is difficult to precisely measure the depth from the skin till the genioglossus. The mean depth may be 25-30 mm. However, the depth must have considerable interindivdual differences. Further, subcutaneous fat in the submandibular region must influence on the depth. Computed tomography, if applicable, would provide important information concerning measure from submental skin to the genioglossus. I added the sentences in the text. If the patient can reproduce dystonic movement (protrusion), correct placement of the tip of the needle in the genioglossus (not in the geniohyoid) can be verified easily. I added the sentences in the text.
Reviewer 3 Report
This is an impressive series of lingual dystonia patients. As the authors point out, given the rarity of this condition, having a single center analysis of so many patients with a clear clinical history is quite unique. I would suggest some editing/additions to enhance the manuscript.
- Editing for grammar and typographic errors would help. E.g. Line 82 "patients" is repeated twice. Line 126 "efficiency" should likely be "efficacy." These are just two examples.
- Line 235 is unclear. "Because of 236 these variabilities each author used identical injection method for their own patients since a single 237 approach cannot be effective for all patients." Does this mean that each author used the same injection pattern for their patients? Or did the authors change the pattern based on the patient's examination? The word "identical" is confusing.
- Rates of non-lingual muscle injections need to be reported. The manuscript references additional muscles (masseter, pterygoid) which were injected based on the patient's examination. This has a significant impact on rates of improvement and complications. Reporting rates of response for different subtypes of lingual dystonia are not useful unless the reader knows to what extent additional muscles are being injected.
Author Response
- Editing for grammar and typographic errors would help. E.g. Line 82 "patients" is repeated twice. Line 126 "efficiency" should likely be "efficacy." These are just two examples.
Thank you for your comment. I corrected the errors. Furthermore, I have throughout my manuscript corrected by a professional English editing service.
- Line 235 is unclear. "Because of 236 these variabilities each author used identical injection method for their own patients since a single 237 approach cannot be effective for all patients." Does this mean that each author used the same injection pattern for their patients? Or did the authors change the pattern based on the patient's examination? The word "identical" is confusing.
I rewrote the sentence as follows. “In light of this variability, the same method of approach is unlikely to be effective for all patients. In spite of this, researchers used the same injection method for all patients.”
- Rates of non-lingual muscle injections need to be reported. The manuscript references additional muscles (masseter, pterygoid) which were injected based on the patient's examination. This has a significant impact on rates of improvement and complications. Reporting rates of response for different subtypes of lingual dystonia are not useful unless the reader knows to what extent additional muscles are being injected.
Thank you very much for valuable comments. I calculated results of non-lingual muscle injections. I added the following sentence in the Results. “Depending on the symptoms of each patient, injected muscles, other than extrinsic and intrinsic muscles, included the lateral pterygoid (24 patients), masseter (23 patients), posterior belly of the digastric (8 patients), medial pterygoid (5 patients), temporalis (4 patients), geniohyoid (4 patients), anterior belly of the digastric (3 patients), sternocleidomastoid (3 patients), orbicularis oris (3 patients), risorius (2 patients), mentalis (1 patient), and buccinator (1 patient) muscles.”
Round 2
Reviewer 1 Report
The ms in the revised form is a massive step forward. Now I have no further sugggestions to improve the ms.